# pH and Redox Dual-Responsive Mesoporous Silica Nanoparticle as Nanovehicle for Improving Fungicidal Efficiency

**DOI:** 10.3390/ma15062207

**Published:** 2022-03-17

**Authors:** Litao Wu, Hua Pan, Weilan Huang, Zhongxuan Hu, Meijing Wang, Fang Zhang

**Affiliations:** Faculty of Environment and Life, Beijing University of Technology, Beijing 100124, China; wltuiao@163.com (L.W.); panh@emails.bjut.edu.cn (H.P.); huangweilan1226@163.com (W.H.); layernano@emails.bjut.edu.cn (Z.H.); meijingwang131@163.com (M.W.)

**Keywords:** bimodal mesoporous silica, Prochloraz, pH/redox dual-responsive, controlled release, biosafety

## Abstract

Prochloraz (Pro) controlled-release nanoparticles (NPs) based on bimodal mesoporous silica (BMMs) with redox and pH dual responses were successfully prepared in this study. BMMs was modified by a silane coupling agent containing a disulfide bond, and β-cyclodextrin (β-CD) was grafted on the surface of the NPs through host–guest interaction. Pro was encapsulated into the pores of nanoparticles by physical adsorption. NPs had a spherical structure, and their average diameter was 546.4 ± 3.0 nm as measured by dynamic light scattering. The loading rate of Pro was 28.3%, and it achieved excellent pH/redox dual-responsive release performance under acidic conditions. Foliage adhesion tests on tomato leaves showed that the NPs had good adhesion properties compared to the commercial formulation. Owing to the protection of the nanocarrier, NPs became more stable under ultraviolet light and high temperature, which improves the efficient utilization of Pro. Biological activity tests showed that the NPs exhibited effective antifungal activity, and the benign biosafety of the nanocarrier was also observed through toxicology tests on cell viability and the growth of *Escherichia*
*coli* (*E. coli*). This work provides a promising approach to improving the efficient utilization of pesticides and reducing environmental pollution.

## 1. Introduction

Pesticides, which are widely used in modern agriculture, play an irreplaceable role in crop disease control and enhancing grain yield [1,2]. However, the effective utilization rate of traditionally formulated pesticides in field applications is low due to photolysis, biodegradation, volatilization, and rain erosion, among other factors, which seriously contaminate ecosystems [3,4]. Therefore, improving the utilization of pesticides has been a crucial issue over the past decade. Recently, with the rapid development and application of nanotechnology in the agricultural field, a series of environmental-stimulus-responsive pesticide nanosystems [5,6,7], such as pH [8,9,10], temperature [11,12,13,14], enzyme [15,16,17], redox [18,19,20], and light-responsive materials [21,22,23], were developed to improve the utilization rate of pesticides and reduce environmental risk. However, to the best of our knowledge, few reports on dual-responsive pesticide delivery nanosystems exist.

Glutathione (GSH), an important antioxidant, widely exists in various animals, plants, fungi, and bacteria because of its high antioxidant performance [24,25]. The reducing hydrogen provided by the sulfhydryl group (–SH–) in glutathione breaks the disulfide bond (–S–S–), generates glutathione oxidized (GSSG) by itself, and reduces the disulfide bond to a sulfhydryl group. The integration of –S–S– bonds can endow nanoparticles (NPs) with redox-responsive release due to the breakage of –S–S– bonds in the presence of glutathione [26,27]. β-Cyclodextrin (β-CD) is widely used because of its nontoxicity and unique molecular capsule structure [28]. β-CD contains a large number of hydroxyl groups and a unique cavity structure, and so can recognize and interact with guest molecules through hydrogen bonds and the van der Waals force to form host–guest inclusion complexes. However, β-CD is not stable, and the connection between β-CD and guest molecules is broken by H^+^, resulting in a “decapping” effect under acidic environments [29,30]. A smart dual-responsive nanopesticide delivery system, however, can be designed to improve the efficient utilization of pesticides based on the GSH contained in fungi and the acidification of an environment by fungi during colonization [19,31]. Compared with traditional mesoporous materials, BMMs is a kind of mesoporous silica material with a wormlike pore (3 nm) and a spherical particle-stacking hole (10–30 nm) in the double-channel structure [32,33,34]. BMMs are widely used as nanocarriers due to their tunable structure, high loading capacity, good stability, and environmental friendliness. In addition, the uptake and translocation of nanocarriers in plants and fungus mycelia were observed owing to the 20–50 nm size of BMMs. *Rhizoctonia solani* (*R. solani*), as a widespread soilborne plant pathogenic fungus, infects many economically important agricultural and horticultural crops, and results in a 20–40% global crop-yield loss. The main approach to control *R. solani* depends on fungicide sprays [35,36,37]. Prochloraz (Pro) is a broad-spectrum imidazole fungicide that is applied as protection against crop diseases caused by fungi, including *R. solani* [15,38,39]. However, the utilization of Pro is limited due to its poor light instability and short effective period. To resolve these issues, some smart delivery nanosystems, such as organic polymers [40], mesoporous silica [17,39] and metal–organic frameworks [31], were developed to improve the stability and utilization rate of Pro. These nanosystems are effective in monitoring the influence of Pro on the environment.

In this study, we grafted a disulfide-modified silane coupling agent onto the surface of BMMs, and Pro was loaded in NPs by physical adsorption. The pyridine ring (Py) on the surface of the silane coupling agent was recognized by β-CD to form a host–guest supramolecular valve, which was further wrapped on the BMMs carrier to form a pH and redox dual-responsive pesticide release system (Pro@BMMs-SS-Py/β-CD). The fabrication process is shown in Figure 1. The physicochemical properties, release behavior, stability, adhesion, and bioactivity of the NPs were also investigated. This study provides a novel strategy for managing fungal disease and reducing environmental pollution.

## 2. Materials and Methods

### 2.1. Materials

Prochloraz technical (Pro TC, 97.5%) was obtained from Beijing Mindleader Agroscience Co., Ltd. (Beijing, China). Methacrylic acid (MAA, AR) was purchased from Tianjin Fuchen Chemical Reagent Co., Ltd. (Tianjin, China). Cetyl trimethyl ammonium bromide (CTAB, 98%), tetraethyl orthosilicate (TEOS), acetic acid (99.8%), potassium bromide (KBr), and other organic solvents were purchased from Sinopharm Chemical Reagent Beijing Co., Ltd. (Beijing, China). N-hexane was purchased from the Beijing Chemical Plant (Beijing, China). β-cyclodextrin (β-CD) was supplied by Sigma Aldrich (Saint Louis, MO, USA). 2,2′-Dithiodipyridine (Aldrithiol, 98%) was purchased from Shanghai Macklin Biochemical Technology Co., Ltd. (Shanghai, China). 3-Mercaptopropyltrimethoxysilane (MPTES, 97%) was obtained from Beijing Bailingwei Technology Co., Ltd. (Beijing, China). Dialysis membrane (molecular weight cut-off 3500 Da) was purchased from Beijing Kebiquan Biotechnology Co., Ltd. (Beijing, China). Deionized water (18 MΩ cm^−1^) was prepared using a Milli-Q water purification system (Millipore, Milford, MA, USA). *Rhizoctonia solani* (*R. solani*, bio-53510) was provided by the Institute of Chinese Academy of Agricultural Sciences.

### 2.2. Preparation of BMMs

BMMs nanoparticles were synthesized with the sol–gel method. A total of 1.0448 g of CTAB was dissolved in 41.6 mL of deionized water and stirred continuously. Afterwards, 3.2 mL of TEOS was added dropwise, and 0.96 mL of ammonium hydroxide was added quickly until the solution changed into white gel. The white gel was filtered and washed, followed by drying for 6 h at 120 °C to obtain raw nanomaterial powder. The powder was heated to 550 °C at a rate of 5 °C min^−1^ for 5 h to remove CTAB and yield BMMs.

### 2.3. Preparation of BMMs-SS-Py NPs

First, 1 g of MPTES was dissolved in 5 mL of dichloromethane at 25 °C, and the solution was added dropwise to a solution of 2,2’-dithiopyridine (3.36 g) for 2 h. The solution was stirred continuously for 1 h and then extracted with petroleum ether. After removing the solvent in vacuum, product [MPTES-SS-Py, (CH_3_O)_3_Si-(CH_2_)_3_-SS-C_5_H_5_N] was obtained.

BMMs (100 mg) was activated for 2 h at 120 °C and dispersed in 20 mL of toluene. We added 50 μL of MPTES-SS-Py into the solution, stirred continuously at 80 °C in a nitrogen environment for 4 h, washed with toluene three times, centrifuged at 6500 rpm for 6 min, and then dried at 50 °C for 8 h.

### 2.4. Preparation of Pro@BMMs-SS-Py NPs

BMMs-SS-Py (100 mg) and Prochloraz (200 mg, in 40 mL of N-hexane) were added to a flask, and the solution was stirred at 25 °C for 24 h, centrifuged at 6500 rpm, washed with N-hexane three times, and dried at 45 °C for 8 h. The obtained product was denoted as Pro@BMMs-SS-Py.

### 2.5. Preparation of Pro@BMMs-SS-Py/β-CD NPs

Pro@BMMs-SS-Py (100 mg) was dispersed in phosphate buffered saline (PBS, pH = 7.4), followed by the addition of 50 mg of β-CD. The mixture was stirred for 24 h. Next, the solution was rinsed with PBS three times, centrifuged for 8 min at 6500 rpm, and dried at 45 °C for 8 h to obtain Pro@BMMs-SS-Py/β-CD NPs.

### 2.6. Characterization of Pro@BMMs-SS-Py/β-CD NPs

The morphologies of the obtained NPs were observed by scanning electron microscopy (SEM) (Zeiss, Crossbeam 350/550, Oberkochen, Germany). Synthesized NPs were suspended in deionized water at pH 7.0 ± 0.05. Structural and interaction analyses of products were carried out using X-ray powder diffraction (XRPD, D8 ADV ANCE X, Bruker/AXS, Inc., Karlsruhe, Germany) and Fourier transform infrared spectroscopy (FTIR, Nicolet Nexus 470, Nicolet Instrument Corp., Concord, CA, USA). Binding affinity among BMMs, MPTES-SS-Py, and β-CD was measured using an ultraviolet–visible-light (UV–vis) spectrophotometer (UV-2600, Shimadzu Co., Ltd., Tokyo, Japan) at a wavelength of 220–380 nm. The particle size, polymer dispersity index (PDI), and zeta potential of BMMs, BMMs-SS-Py, Pro@BMMs-SS-Py, and Pro@BMMs-SS-Py/β-CD NPs were determined according to dynamic light scattering (DLS) using a nanoparticle analyzer (Zetasizer Nano ZS, Malvern Instruments Ltd., Malvern, UK). The surface areas, pore-size distributions, and pore volumes of the samples were determined by an Autosorb-iQ pore analyzer (Quantachrome, Boynton Beach, FL, USA) and calculated by the Brunauer–Emmett–Teller (BET, Micromeritics, Atlanta, GA, USA) and Barrett–Joyner–Halenda methods. Thermogravimetric analysis (TGA) was conducted with a thermogravimetric analyzer (PerkinElmer, Waltham, MA, USA). To verify the characterization of MPTES-SS-Py, liquid nuclear-magnetic-resonance (NMR) spectra were recorded on an Ascend^TM^ 400 (AVANCE HD III) spectrometer (Bruker, Germany). All chemical shifts were measured relative to residual ^13^C NMR resonances in the deuterated solvents: CD_2_Cl_2_, δ 53.8 ppm for ^13^C.

### 2.7. Pro Measurement

Pro@BMMs-SS-Py/β-CD NPs (5 mg) was dispersed in 30 mL of methanol and lysed on ice by sonication (80 × 30 s with 1-minintervals, Misonix Sonicator 3000, Misonix Inc., NY, USA). Then, 1 mL of the solution was collected and centrifuged at 12,000 rpm for 15 min. Subsequently, Pro content was detected with high-performance liquid chromatography (HPLC, Agilent 1200 Series, Agilent Technologies, Wilmington, DE, USA). The operating parameters for HPLC determination were as follows: C18 reversed-phase column, 5 μm × 4.6 mm × 250 mm; column temperature, 30 °C; detection wavelength, 220 nm; mobile phase, acetonitrile/0.1% acetic acid water (*v*/*v*, 70:30); injection volume, 10 μL; flow rate, 1.0 mL min^−1^.

The loading content (%) of Pro in NPs was calculated as follows: loading content (%) = (weight of Pro encapsulated in Pro@BMMs-SS-Py/β-CD/weight of Pro@BMMs-SS-Py/β-CD) × 100%.

### 2.8. pH/Redox Dual-Responsive Release

Pro@BMMs-SS-Py/β-CD NPs (5 mg) were separately dispersed in 3.0 mL of deionized water containing 0.1% Tween-80 emulsifier in dialysis bags (molecular weight cutoff: 3500 Da). Subsequently, the sealed dialysis bags were placed in 48 mL of release medium at 27 °C in the dark and stirred at 100 rpm. Within certain predetermined time intervals, 1.0 mL of the mixture was removed and an equal volume of fresh solvent added. Pro content was analyzed with HPLC.

To investigate the effects of pH, GSH, and temperature on the release behaviors of Pro from nanoparticles, the contents of Pro were determined by HPLC at different pH levels (4.0, 7.0, and 10.0), GSH (0, 1.0, and 2.0 mM), and temperature (4, 25, and 54 °C). The accumulative Pro release was calculated as follows:$$Q\left(\%\right)=\frac{V_{e}\sum_{n}^{n-1}C_{i}+V_{0}C_{n}}{m}\times 100\%$$
where *Q* is the cumulative release (%) of Pro from the Pro@BMMs-SS-Py/β-CD NPs, *V_e_* is the volume of the release medium taken at a given time interval, *V*_0_ is the volume of release solution; *C_n_* (mg·mL^−1^) is the Pro concentration in the release medium at time n, and *m* (mg) is the total pesticide loaded in the Pro@BMMs-SS-Py/β-CD NPs.

To more systematically elucidate the effect of pH and GSH on the behavior of sustained Pro release from Pro@BMMs-SS-Py/β-CD NPs, the release kinetics of Pro from the NPs was separately investigated using four dynamic mathematical models, namely, a zero-order equation, a first-order equation, the Higuchi model, and the Ritger–Peppas model.

### 2.9. Stability Study

To determine the ultraviolet (UV) light stability of Pro@BMMs-SS-Py/β-CD NPs, 5 mg of Pro@BMMs-SS-Py/β-CD NPs was dispersed in 50 mL of 0.1% Tween-80 aqueous solution and exposed to a 36 W UV lamp (254 nm) at a distance of 20 cm at room temperature. Technical Pro was used as a control. Samples were collected at predetermined time points and analyzed with HPLC.

The storage stability of Pro@BMMs-SS-Py/β-CD NPs was determined according to a previous report [9]. NPs were stored at 4, 25 ± 2, and 54 ± 2 °C for 14 days, and Pro content was analyzed using HPLC. 

### 2.10. Foliage Adhesion Test

Fresh tomato leaves were cultivated in a greenhouse, and leaves were cleaned with deionized water without destroying the leaf structure and then completely air-dried. The dynamic contact angles of the prepared NPs were measured using a contact-angle meter (Kruss DSA 100, JC2000D2M, Powereach Ltd., Shanghai, China). A droplet (2 μL) of the solution was dropped onto the surface of the leaves with a microsyringe. The contact angles were then measured for 1 min. Technical Pro, BMMs, and deionized water were used as controls.

### 2.11. Bioefficacy Test

The fungicidal efficacy of Pro@BMMs-SS-Py/β-CD NPs against *R. solani* was determined using the poison plate method. Potato dextrose agar (PDA) plates were treated with Pro@BMMs-SS-Py/β-CD NPs and Pro technical control (TC) of different concentrations (0, 0.0625, 0.125, 0.25, 0.5, and 1 mg·L^−1^). Deionized water was used as a negative control. The mycelial discs (5 mm in diameter) of *R. solani* were inoculated on the plates for 12 days (d). After incubation at 27 °C for 12 d, the colony diameter of the mycelium was measured by the crisscross method, and 50% inhibition concentration (IC50) of NPs on *R. solani* was calculated. The bioefficacy test of NPs was evaluated by inhibitive rate and IC50 value. All tests were carried out in triplicate.

### 2.12. Biosafety Evaluation

Human bronchial epithelial (16HBE) cells in the logarithmic growth stage were seeded in 96-well plates in triplicate and cultured in 1640 medium supplemented with 10% FBS and 1% penicillin/streptomycin at 37 °C for 24 h. Then, 16HBE cells were treated with different concentrations (0, 31.25, 62.5, 125, and 250 mg·L^−1^) of BMMs-SS-Py/β-CD NPs for 24 h, followed by analysis of cell viability using a cell-counting kit (CCK8, Dojindo, Japan). *Escherichia coli* was cultured in Luria–Bertani (LB) culture medium with different concentrations (0, 31.25, 62.5, 125, and 250 mg·L^−1^) of BMMs-SS-Py/β-CD NPs at 37 °C for 24 h, and *E. coli* concentrations were measured with UV–vis absorptiometry using a photometer (BioPhotometer Plus Model 6132, Eppendorf, Hamburg, Germany) at 600 nm.

### 2.13. Data Analysis

Data were analyzed using Statistical Product and Service Solutions (SPSS 20.0) statistical analysis software (SPSS, Chicago, IL, USA). All experiments were performed in triplicate. Statistical significance was determined as *p* < 0.05.

## 3. Results and Discussion

### 3.1. Characterization and Interaction Analysis

The morphological images of BMMs and Pro@BMMs-SS-Py/β-CD NPs, observed by SEM, are shown in Figure 2. The nanoscale spherical morphology of Pro@BMMs-SS-Py/β-CD NPs was similar to that of BMMs, indicating that MPTES-SS-Py grafting, Pro loading, and encapsulation with β-CD did not destroy BMMs morphology. After Pro had been loaded and the mesopore surface grafted with β-CD, the particle size of NPs increased from 387.2 ± 3.8 nm (BMMs) to 546.4 ± 3.0 nm (Pro@BMMs-SS-Py/β-CD) (Figure 3A). The PDI value of Pro@BMMs-SS-Py/β-CD NPs was 0.32 ± 0.02, which was lower than that of the other NPs, showing that the nanomaterials could be steadily dispersed in water. As shown in Figure 3B, the zeta potential of BMMs was −15.27 ± 0.25 mV owing to the presence of –OH on the surface of the mesoporous silica. After the grafting of MPTES-SS-Py, the zeta-potential value of BMMs decreased to −18.73 ± 0.41 mV because it was easy for the nucleophilic substitution reaction to take place at the pyridine rings contained in the MPTES-SS-Py compound. After loading Pro, the zeta-potential value of Pro@BMMs-SS-Py NPs increased to −17.5 ± 0.3 mV due to the positive charge of Pro [31], indicating that the Pro pesticide was successfully loaded into the NPs. After the modification of β-CD, the zeta-potential value increased to −15.9 ± 0.5 mV because the reaction between the pyridine ring and water was inhibited by introduction of β-CD.

The crystal structure of NPs was also tested by XRPD, and results are shown in Figure 3C. BMMs had an obvious (100) crystal-plane diffraction peak at 2θ = 1.86°, indicating that they had a highly ordered double-hole structure. After the metafiction of MPTES-SS-Py, an identical crystal-plane diffraction peak (100) was observed, indicating that BMMs-SS-Py NPs still maintained an ordered mesoporous structure; in addition, peak intensity increased to 1.95°, and d value decreased from 47.34 to 44.71 nm because the successful grafting of MPTES-SS-Py and the doping of new atoms in the silicon wall led to a decrease in the lattice constant of BMMs. After Pro had been loaded into BMMs, the strength of the XRPD peaks (2θ = 1.97°) of Pro@BMMs-SS-Py NPs significantly decreased, and the shape broadened, showing that the mesoporous structure was significantly affected. The peak value of NPs decreased further after grafting β-CD, indicating that β-CD was successfully encapsulated into the system.

The FTIR spectra of BMMs, BMMs-SS-Py, Pro@BMMs-SS-Py, and Pro@BMMs-SS-Py/β-CD were determined to evaluate the NPs structural changes with different functional groups (Figure 3D). BMMs exhibited characteristic peaks at 1086 and 811 cm^−1^, which were the antisymmetrical and symmetrical stretching-vibration peaks of Si–O–Si groups, respectively. The absorption band at 1648 cm^−1^ was the specific stretching vibration of the pyridine-ring skeleton, indicating that MPTES-SS-Py was successfully grafted onto the BMMs surface. The characteristic absorption peak of Pro at 1469 and 1742 cm^−1^ suggested that the Pro pesticide was adsorbed in the mesoporous silica. The porosity characterizations of NPs were investigated using the N_2_ adsorption-desorption technique. As shown in Figure 4A, the N_2_ adsorption-desorption isotherms of BMMs, BMMs-SS-Py, Pro@BMMs-SS-Py, and Pro@BMMs-SS-Py/β-CD belonged to the Langmuir IV isotherm with two hysteresis loops. The first hysteresis loop, at 0.3 < P/P_0_ < 0.5, increased rapidly owing to the monolayer adsorption of nitrogen. The second hysteresis loop appeared at P/P_0_ = 0.8–0.95, indicating that the capillary tube of the particle accumulation pore had been condensed. The corresponding pore-size distribution revealed that NPs had a dual-model structure and two pore sizes (Figure 4B). After modification of MPTES-SS-Py in BMMs, the shape of the adsorption isotherm remained basically unchanged compared with that of BMMs. After loading Pro, the BET specific surface area and pore volume of Pro@BMMs-SS-Py decreased significantly to 56.15 and 0.25, respectively, implying that Pro molecules occupied pore channels of the NPs, and were successfully loaded into Pro@BMMs-SS-Py/β-CD NPs. However, the pore volume and specific surface area of NPs increased slightly after grafting of β-CD, which might be due to the slight leakage of the Pro when grafting β-CD (Table 1).

To verify that β-CD could self-assemble with BMMs-SS-Py to form a gatekeeper for the controlled-release of pesticide Pro, we separately detected the UV absorption spectra of NPs under UV irradiation (Figure 4C). BMMs-SS-Py produced an obvious UV absorption peak at 280 nm due to the n–π transition by the pyridine ring, which indicated that MPTES-SS-Py was successfully grafted onto the BMMs surface. When β-CD had been coated onto the surface of the NPs, the pyridine ring was covered and interfered with by the high electron cloud in the β-CD cavity, which hindered the source and led to the red shift of the spectrum.

The loading ratio of Pro@BMMs-SS-Py/β-CD NPs was determined with TGA (Figure 4D).

Significant weight loss of NPs occurred between 150 and 800 °C. Weight loss before 150 °C was caused by the gasification of water and residual solvent contained in NPs. The TG curve of Pro@BMMs-SS-Py/β-CD NPs obviously showed two levels of decrease. The first weight-loss peak occurred at 150–300 °C, which was mainly due to the decomposition of Pro in the NPs channel. The second obvious weight-loss interval appeared at 300–500 °C due to the decomposition of MPTES-SS-Py (15.2%) and β-CD (6%) grafted onto the surface of BMMs. The loading rate of Pro in Pro@BMMs-SS-Py/β-CD NPs was about 28.3%, which was largely consistent with results measured with HPLC (28.1%). It was thus proved that Pro was successfully loaded into BMMs-SS-Py/β-CD NPs.

The success of MPTES-SS-Py synthesis was also verified by nuclear magnetic carbon spectroscopy (Figure A1). Four signals were displayed in the 0–50 ppm region: resonances at approximately 42 and 23 ppm were attributed to the carbon atoms of a propyl moiety situated in α and β positions of the disulfide bond, the resonance at approximately 10 ppm to the carbon atom directly connected to the silicon atom, and the 48 ppm resonance to a methoxy group. In addition, signals from all carbon atoms in the pyridine base appeared at 120–150 ppm.

### 3.2. Foliage Adhesion

Foliage-adhesion experiments were conducted to evaluate the adhesion behavior of Pro@BMMs-SS-Py/β-CD NPs. As shown in Figure 5, the dynamic contact angle of Pro@BMMs-SS-Py/β-CD NPs decreased from 62.89 to 60.11 in 1 min, which was obviously lower than that of Pro technical solution (from 93.87 to 85.09) and deionized water (from 85.81 to 79.23). Data showed that β-CD-coated Pro@BMMs-SS-Py microcapsules exhibited excellent adhesion properties. Compared with deionized water and technical Pro, a large amount of –OH on the β-CD surface interacted with –CHO and –COOH on the blade surface, which increased the electrostatic force between NPs and the blade, and enhanced NPs wettability.

### 3.3. Stability Study

The utilization of Pro was limited due to its light instability. To evaluate the potential effects of UV irradiation on the stability of Pro@BMMs-SS-Py/β-CD NPs, photolytic rate curves at different UV exposure times were obtained (Figure 6A). Compared with technical Pro, the photolysis rate of Pro in NPs was slow, with 72% degradation after 144 h of exposure to continuous UV light. These data indicated that the photolytic stability of Pro in the delivery nanosystem was significantly promoted by the efficient protection of nanocarriers.

Active ingredients of pesticides are easily affected by storage stability. The Pro contents of Pro@BMMs-SS-Py/β-CD NPs at three different temperatures (4, 25, and 54 °C) were measured separately. Figure 6B shows that Pro loading rates during storage at 4 °C (28.1%) and 25 °C (28%) exhibited no obvious changes. A loss of less than 1% of Pro was observed after 14 days at 54 °C because the melting point of Pro is 46.5–49.3 °C, and Pro might undergo degradation during the transition from solid to liquid. These results showed that the encapsulation of the nanocarriers increased Pro’s storage stability.

### 3.4. pH/Redox Dual-Responsive Release Behavior

The release behaviors of Pro@BMMs-SS-Py/β-CD NPs were investigated at different pH values. The cumulative release curves of Pro are shown in Figure 7A. The cumulative-release rate gradually increased as pH value decreased, and the Pro release rate reached 41.43% at 24 h at pH 4. After 120 h, the release rate of Pro increased to 64.92%, and those at pH 7 and pH 10 were only 55.62% and 47.14%, respectively. This was due to the degradation of “gatekeeper” β-CD encapsulated on the surface of nanocarriers under acidic conditions, which promoted the release of Pro. In addition, excessive hydrogen ions (H^+^) in the acid solution were bound to the C = N bonds of the pyridine ring, resulting in the breakage of the connection of β-CD and pyridine rings, and ultimately in Pro release.

The release behaviors of Pro@BMMs-SS-Py/β-CD NPs were further explored at different GSH values. The content of GSH in fungi is approximately 0.5–8 mM. In our previous work, when GSH values were 4 and 8 mM, the Pro release profile was similar to that of GSH at 2 mM. In addition, the grafted GSH concentration had an obvious effect on the pesticide-loading capacity of the nanosystem. Since the disulfide bond accounted for a large surface area of the NPs, a high GSH concentration reduced Pro’s pesticide-loading capacity. Therefore, we chose 0–2 mM GSH to study the effect on Pro release. The cumulative release curves of Pro with various GSH concentrations are shown in Figure 7C. The release rate of Pro was 94.32% at 120 h, which was 69.58% and 17.44% higher than those of GSH at 0 and 1 mM, respectively. Owing to MPTES-SS-Py being grafted onto the surface of BMMs containing disulfide bonds, GSH contained in the release medium broke the disulfide bonds and accelerated Pro release.

The sustained release curves of Pro at different temperatures are shown in Figure 8. The cumulative release rate of Pro gradually improved with the increase in temperature. The release ratio of Pro from Pro@BMMs-SS-Py/β-CD NPs at 54 °C was 55.4%, and those at 25 and 4 °C were 48.2% and 46.1%, respectively. This was mainly due to the high temperature accelerating the movement of Pro molecules in the NPs channel [41]. In addition, the melting point of Pro is below 50 °C, and the morphology of Pro encapsulated in the pores of the nanocarrier gradually changed from solid to liquid, which promoted the enhancement of the release rate of NPs.

### 3.5. Release-Kinetics Analysis

To further elucidate the effect of pH and GSH on the release of Pro in Pro@BMMs-SS-Py/β-CD NPs, we studied the release kinetics using Ritger-Peppas kinetics (Figure 7B,D) and the zero- and first-order Higuchi models (Figure A2). The regression-coefficient (R^2^) values of the Ritger-Peppas kinetic equation were higher than those of the three other mathematical models under the conditions of different pH values and GSH concentrations, indicating that the Ritger-Peppas kinetic model was more suitable for studying the Pro release behavior (Table A1 and Table A2). The values of n were lower than 0.45, proving that the Pro release in Pro@BMMs-SS-Py/β-CD mainly followed Fick diffusion.

### 3.6. Bioactivity Test

The antifungal activity of Pro@BMMs-SS-Py/β-CD NPs against *R. solani* was investigated using the growth-rate method. As shown in Figure 9, the inhibition rates of NPs and Pro TC were 85.5% and 73.5%, respectively, at the Pro-as-an-active-ingredient concentration of 1 mg·L^−1^ after 12 days. Accordingly, the IC50 values of Pro@BMMs-SS-Py/β-CD NPs (0.2003 ± 0.0018) were 11% lower than those of Pro TC (0.2249 ± 0.02) owing to the sustained release of Pro in the nanocomplex systems. *R. solani* secreted acidic substances during colonization [42], and acidification of the external environment caused the pH response of the NPs. As nanocarriers were modified by β-CD, the connection between β-CD and guest molecules was broken by H^+^, resulting in the release of Pro under acidic environments. In addition, the presence of glutathione in the internal environment of fungi caused the redox response of the NPs. The sulfhydryl group (−SH−) in glutathione broken the disulfide bond (−S−S−) integrated in Pro@BMMs-SS-Py/β-CD NPs, and promoted the release of Pro from NPs. The above redox and pH dual response of NPs resulted in the sustained release of Pro from NPs and significantly inhibited the growth of *R. solani*.

### 3.7. Biosafety Evaluation

The biosafety of NPs is a major influencing factor in their application. To further evaluate the biological safety of nanocarriers, the toxicological effects of different concentrations of BMMs-SS-Py/β-CD NPs on 16HBE cells and *E. coli* were studied. Figure 10 shows that different concentrations of BMMs-SS-Py/β-CD NPs had little influence on the growth of 16HBE cells and *E. coli*. NPs also promoted the growth of *E. coli* with increasing NPs concentration, which might have been due to the introduction of β-CD molecules. As a result, this dual-responsive nanocarrier exhibited excellent biological safety.

## 4. Conclusions

In this work, we prepared a novel pesticide delivery nanosystem with both pH and redox dual response by the sol-gel method. Pro@BMMs-SS-Py/β-CD NPs had a uniformly spherical morphology and good dispersibility in water. The nanocomplex showed excellent pH and redox dual-responsive controlled-release performance owing to the host-guest complex between the MPTES-SS-Py and β-CD. The Ritger-Peppas kinetic model was fitted with the release behavior of Pro. The nanocarrier displayed good adhesion on leaf surfaces, and benign stability for light and temperature. The sustained fungicidal efficacy against *Rhizoctonia solani* indicated that Pro@BMMs-SS-Py/β-CD NPs could effectively improve the efficacy of Pro and reduce pesticide residue. Moreover, the NPs possessed excellent biosafety. Some studies on Pro sustained release systems were performed, but the nanomaterials and the release mechanisms of these delivery nanosystems are different. As shown in Table 2, Pro nanoparticles display higher stability and better antifungal activity than that of traditional Pro in different stimuli-responsive environments, indicating that nanocarriers are highly effective in the protection of pesticides. We systematically explored the stability, adhesion properties, and bioactivity of Pro@BMMs-SS-Py/β-CD NPs under redox/pH dual-responsive stimuli, which provided a novel approach for improving the effective utilization of Pro. Therefore, this work provides a promising strategy to decreasing the risks to the environment, and can promote the development of green agriculture.

## Figures and Tables

**Figure 1 materials-15-02207-f001:**
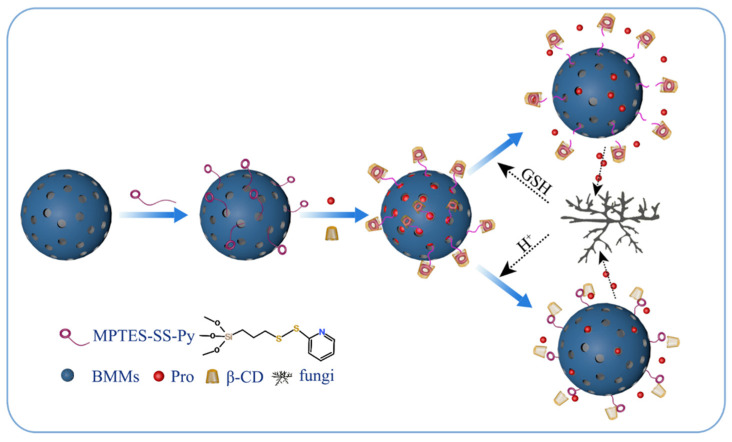
Schematic diagram for preparation of Pro@BMMs-SS-Py/β-CD nanoparticles and their dual-responsive release mechanism.

**Figure 2 materials-15-02207-f002:**
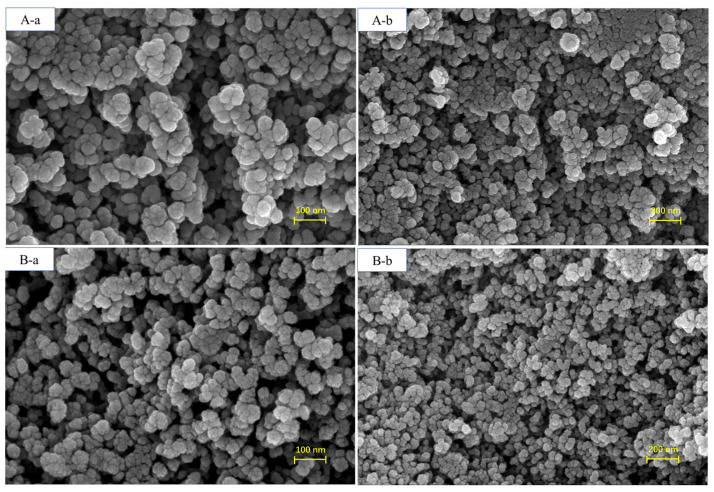
SEM images of (**A**-**a**, **A**-**b**) BMMs and (**B**-**a**, **B**-**b**) Pro@BMMs-SS-Py/β-CD nanoparticles.

**Figure 3 materials-15-02207-f003:**
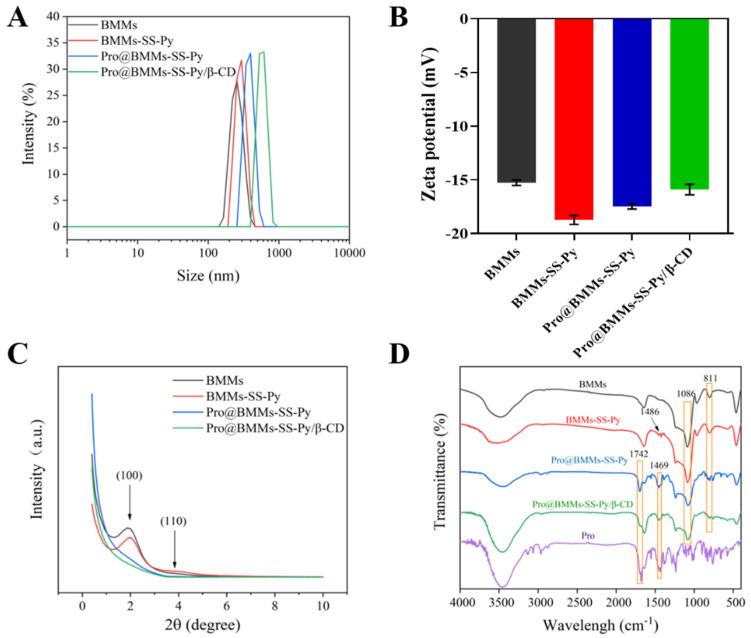
(**A**) Particle size distributions (**B**), zeta potentials, (**C**) XRPD patterns, and (**D**) FTIR spectra of Pro@BMMs-SS-Py/β-CD NPs and control samples.

**Figure 4 materials-15-02207-f004:**
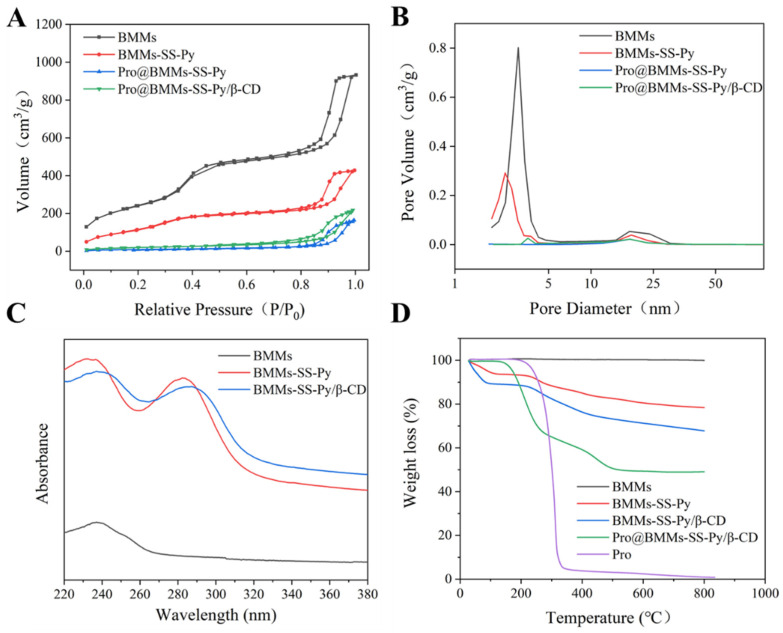
(**A**) BET isotherms, (**B**) pore size distribution curves, (**C**) UV–vis spectra, and (**D**) TGA curves of Pro@BMMs-SS-Py/β-CD NPs and control samples.

**Figure 5 materials-15-02207-f005:**
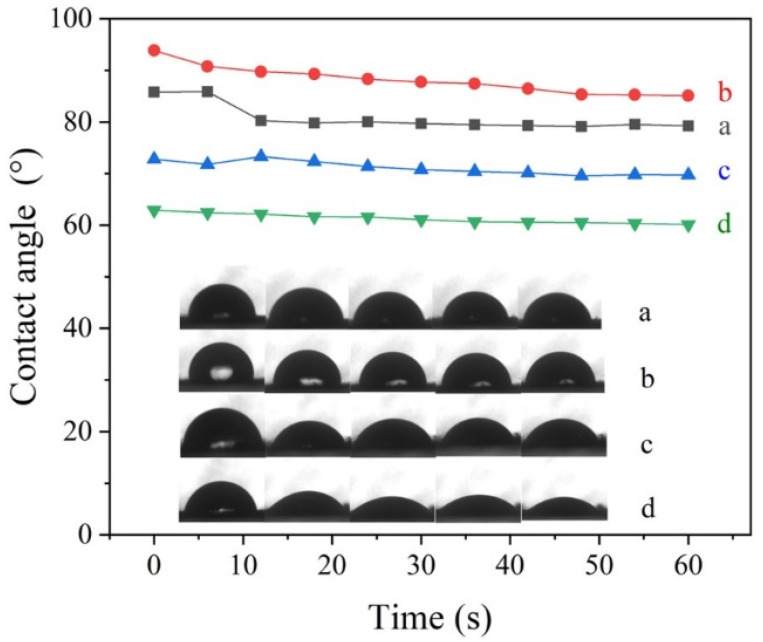
Contact angle values and images of (**a**) deionized water, (**b**) Pro TC, (**c**) BMMs and (**d**) Pro@BMMs-SS-Py/β-CD NPs on tomato leaves.

**Figure 6 materials-15-02207-f006:**
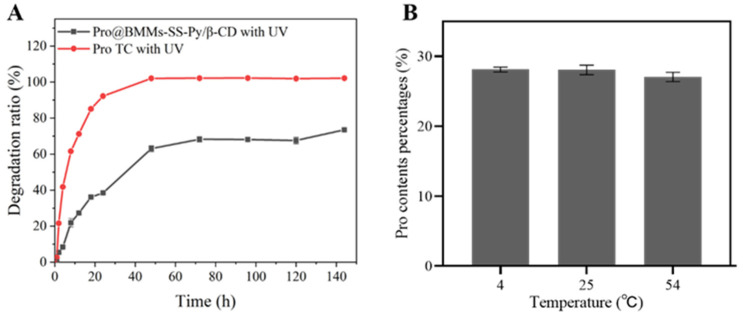
(**A**) Photostability and (**B**) storage stability of Pro@BMMs-SS-Py/β-CD NPs.

**Figure 7 materials-15-02207-f007:**
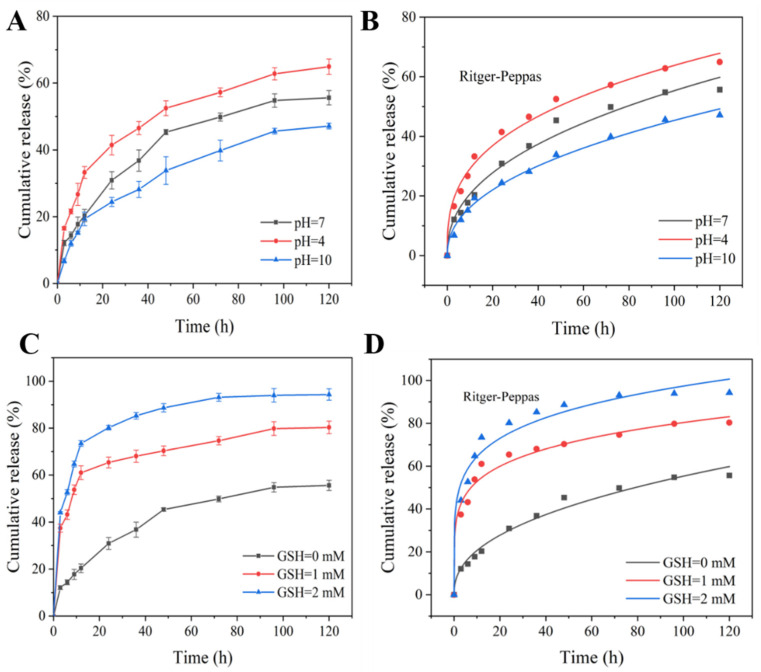
Cumulative release curves and Ritger-Peppas models of Pro@BMMs-SS-Py/β-CD NPs with different (**A**,**B**) pH and (**C**,**D**) GSH values.

**Figure 8 materials-15-02207-f008:**
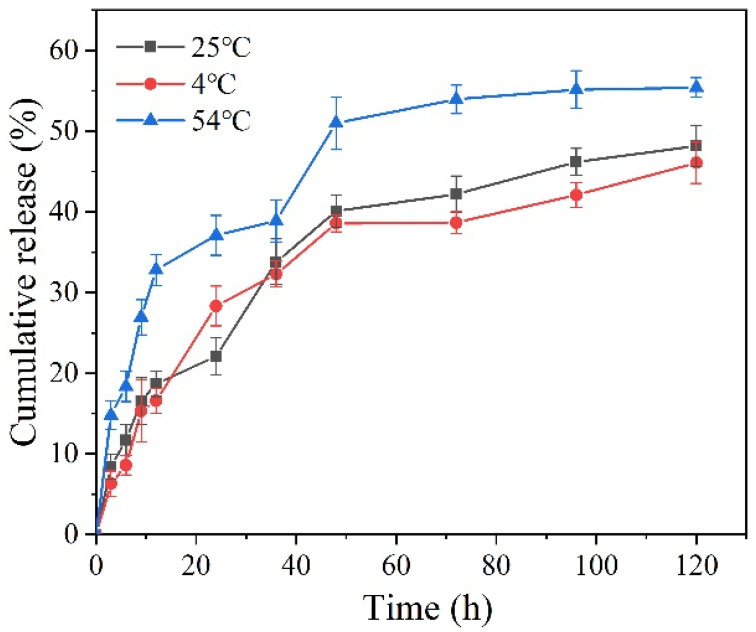
Cumulative release curves of Pro@BMMs-SS-Py/β-CD NPs at different temperatures.

**Figure 9 materials-15-02207-f009:**
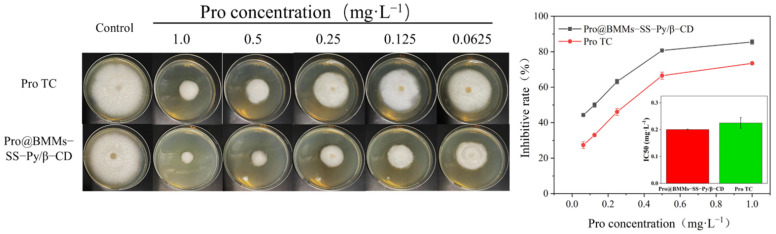
Fungicidal activity of Pro@BMMs−SS−Py/β−CD NPs against *R**. solani*.

**Figure 10 materials-15-02207-f010:**
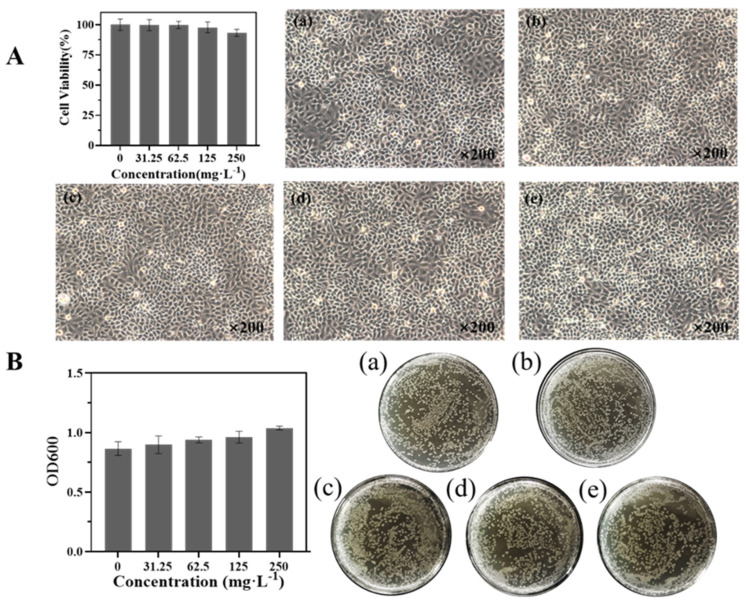
Biosafety evaluation of BMMs−SS−Py/β−CD NPs against (**A**) 16HBE cells and (**B**) *E. coli*. The concentrations of Nps in 16HBE cells and *E. coli.* (**a**) 0 mg·L^−1^, (**b**) 31.25 mg·L^−1^, (**c**) 62.5 mg·L^−1^, (**d**) 125 mg·L^−1^, and (**e**) 250 mg·L^−1, respectively.^

**Table 1 materials-15-02207-t001:** Mesoporous structure characterization, size, and zeta potential of samples.

Sample	S_BET_ (m^2^/g)	V_t_ (cm^3^/g)	Pore Size (nm)	Size (nm)	PDI	Zeta (mV)
BMMs	1126.57	1.48	5.26	295.3 ± 3.9	0.52 ± 0.02	−15.3 ± 0.3
BMMs-SS-Py	546.93	0.69	5.07	360.6 ± 2.8	0.66 ± 0.04	−18.7 ± 0.4
Pro@BMMs-SS-Py	56.15	0.25	—	466.6 ± 3.5	0.61 ± 0.03	−17.5 ± 0.3
Pro@BMMs-SS-Py/β-CD	98.08	0.34	—	546.4 ± 3.0	0.32 ± 0.02	−15.9 ± 0.5

**Table 2 materials-15-02207-t002:** Summary of nanosystems encapsulated with Prochloraz.

Nanocarrier	Stimuli	Release Time (h)	Stability (Pro Loss)	Adhesion	EC50 (mg·L^−1^)	Reference
Zif-8	Light and pH	36	60.2 ± 4.6% after 24 h	Pro retention 175.6 ± 1.6 μg·cm^2^ (before leaf) 155.6 ± 11.7 μg·cm^2^ (after leaf)	0.122 ± 0.02 (*S. sclerotiorum*)	[31]
MSN	—	96	—	—	0.3058 (*Botrytis cinerea*)	[39]
MON-CaC	pH and reduction	120	Less than 15% after 24 h	—	0.142 (*S. sclerotiorum*)	[19]
BMMs-PMAA/Fe^3+^	pH	144	54.2% after 7 d, good thermal stability	—	0.184 ± 0.013 (*Rhizoctonia solani*)	[43]
MSNs-chitosan	Esterase and pH	720	About 6.3% after 72 h		12.468 (11.274–13.900) at 96 h (zebrafish)	[17]
Silica-Alginate	—	1440	Good stability under different pH, temperature and light	—	—	[44]
mPEG-PLGA	—	384	—	—	Best germicidal efficacy (*Fusarium graminearum*)	[40]

## Data Availability

Not applicable.

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
