# Peer review of "pH and Redox Dual-Responsive Mesoporous Silica Nanoparticle as Nanovehicle for Improving Fungicidal Efficiency"

_materials, 2022, doi:10.3390/ma15062207_

Round 1

Reviewer 1 Report

Comments:
There is great interest in understanding the controlled release formulations of the fungicide prochloraz to improve his stability and utilization rate. The aim of this manuscript was to encapsulate prochloraz using nanoparticles based on silica modified by grafting  β-cyclodextrin via silane coupling agent containing a disulfide bond (Pro@BMMs-SS-Py/β-CD). Novel slow release system of prochloraz from the nanoparticles in dual response of pH and redox were successfully prepared. The physical and chemical characterization of the formulations was achieved by using Fourier Transform Infrared (FTIR) spectroscopy, Scanning Electron Microscopy (SEM), Thermogravimetric analysis (TGA), X-ray powder diffraction, ultraviolet–visible-light (UV-Vis) spectrophotometer etc… The stability, adhesion, and bioactivity of the nanoparticles were also investigated. The release of fungicide was studied under laboratory conditions. The influences of pH and Glutathione on the release rate were determined. In addition, this technique might improve worker safety in the handling of pesticides avoiding their spray use, increase the efficacy of Pro and reduce pesticide residue.

However, there are many details and justifications missing in the manuscript, resulting in the work not ready for publication at the current stage. The following problems need to be addressed:

  1. The introduction of this manuscript lacks information on the existing release systems, the prospect and application of slow release of prochloraz.
  2. - In the "Results and Discussion section": The results are little discussed. The authors should highlight the slow-release effect of prochloraz from Pro@BMMs-SS-Py/β-CD nanoparticles compared with other sustained release systems? Several examples are needed to illustrate the problem, so it should be summarized and presented in a table? - Stability, adhesion and bioactivity results should be compared to the literature.
  3. In Figure 4D, the TGA of Pro@BMMs-SS-Py/β-CD obviously showed two levels of decrease between 200-280 and between 280-400 but this was not mentioned in the manuscript.
  4. In the Preparation of Pro@BMMs-SS-Py/β-CD Nps, was the buffer used "Phosphate-Buffered Saline (PBS) or "phosphate buffer"?
  5. In the results, the authors announced that "the loading ratio of Pro@BMMs-SS-Py/β-CD Nps was determined by TGA" but in the Materials and methods section, the loading content was presented in the Pro measurement subsection (using HPLC measurement). The authors should clarify this point.
  6. Page 5, lines 174 to 176: "The storage stability of Pro@BMMs-SS-Py/β-CD Nps was determined according to a previous report [9]. The Nps were stored at 4°C, 25°C ± 2°C, and 54°C ± 2°C for 14 d, and the Pro content was analyzed using HPLC". How the authors achieve total release of the encapsulated pro for HPLC measurement.
  7. Temperature is an important modulator for the controlled release of pesticides. It would be interesting to study the effect of temperature on the release of pro from nanoparticles.
  8. The prochloraz encapsulation must be clearly stated in the Abstract section.

Reviewer 2 Report

Overall the research was properly conducted and the findings were very good. However, only minor corrections were detected as indicated in the returned manuscript. 

Reviewer 3 Report

1. Introduction is poorly written and is vulgarized. Authors need to add the statistical data for plant disease, major causing fungal genera and potentially reported NPs. Then should be describe the flaws and overcome it with your objectives. 2. Author mentioned that Rhizoctonia solani secreted acidic substances during colonization, and acidification of the external environment caused the pH response of the Nps. Please cite the published studies. 3. Authors claim of "Fungicidal activity of Pro@BMMs-SS-Py/β-CD Nps against Rhizoctonia solani" is completely vague. OR should provide new pics with the zone of inhibition. 4. Results are discussed poorly.

Round 2

Reviewer 3 Report

Accept